# Assessing Individual Performance in Team Sports: A New Method Developed in Youth Volleyball

**DOI:** 10.3390/jfmk4030053

**Published:** 2019-08-04

**Authors:** Elisa Bisagno, Sergio Morra, Martina Basciano, Carola Rosina, Francesca Vitali

**Affiliations:** 1Department of Education and Human Sciences, University of Modena and Reggio Emilia, 42121 Reggio Emilia, Italy; 2Department of Education, University of Genova, 16128 Genova, Italy; 3Department of Neurosciences, Biomedicine, and Movement Sciences, University of Verona, 37131 Verona, Italy

**Keywords:** volleyball players, individual performance, youth sport

## Abstract

Studying the role of individual differences in team sports performance is a challenge. The main problem is having an available measure of individual performance of each member of the team. In particular, in youth sports, where the level of specialization is reactively low, it appears appropriate that this measure takes the entire performance of the athlete into consideration (i.e., that it assesses all of the athlete’s gestures), while maintaining an ecological validity criterion. Therefore, we devised and calculated an individual assessment measure in volleyball following the subsequent steps: Firstly, we video-recorded at least three volleyball games for each of the 114 youth volleyball players who participated in the study. Then, two independent expert observers evaluated each individual performance by attributing a score to every single gesture performed by the athletes during the games. The derived individual score was adjusted and controlled for the team performance measure, namely the result of each Set the athlete participated in (and for the amount of participation of the athlete to each game). The final measure of individual performance in volleyball proved to be reliable, showing a high level of interrater agreement (*r* = .841, *p* < .001) and a significant correlation with the amount of experience in volleyball (*r* = .173, *p* < .05).

## 1. Introduction

While the importance of group interactions in team sports is widely acknowledged as a determinant of the final result of a game [1,2], the individual performance in team sports also plays a substantial role in building performance [3,4]. In sport sciences, some studies investigated the role of specific fundamentals as a function of team success [5]. Others investigated the relationship between specific performance and some individual features of volleyball players, such as lower body strength [6,7]. However, when it comes to evaluating the global performance in an ecological context, finding a way to evaluate the individual in team sports is not an easy task, because it requires quantifying the role of a team member in detail, which is difficult to analyze in a systematic way (i.e., isolating one’s gesture from the surrounding environment) [8]. For this reason, in literature, there is a scarcity of proposals on how to derive an individual performance measure in team sports, such as volleyball.

Indeed, the very essence of team sports lies in their unpredictability, due to the interactions with both team members and opponents. In sport sciences, a classic categorization distinguishes sports according to the degree to which the environment they are played in is predictable or not [9]. Closed-skills sports are generally played in a predictable environment (e.g., shooting, gymnastics, swimming). Open-skills sports are those that occur in highly unpredictable environments, and team sports (e.g., volleyball, basketball, rugby) are in this sense prototypical. Many individual sports involve closed skills, whereas many team sports involve open skills. Nevertheless, a clear classification of sports as typified by open- or closed-skills is difficult; indeed, many sports include a combination of both open and closed skills [10]. With respect to this, Lames & McGarry [11] consider team sports performances as being unique action chains, which are context-and-time-dependent and, thus, not repeatable. From the game’s perspective, this characteristic is the one that makes the discipline endearing, but it also represents a confounding variable if one needs to assess an individual’s performance, because devising an “online” assessment (i.e., a measure to assess a single member of the team on each one of his/her gestures) becomes necessary. In volleyball, a measure of motor learning of the attack gesture was developed by Bisagno & Morra [12], who used a task-analysis procedure to isolate the different components that need to be controlled while performing a gesture. However, assessing motor learning is relatively easier than assessing performance in actual competitions, because there is a fewer amount of intervening variables to control for. For example, in competitions, a determinant is represented by the opponents: against unprepared opponents, scoring is easier, while, against highly skilled teams, being able to score has greater merit, and this aspect needs to be controlled when devising a measure of performance. Moreover, since, in volleyball, not all the players play the same amount of Sets and time, it is therefore important to give more weight to the competitions in which the athlete participates more in the game. This is particularly true in youth volleyball, in which the official team is less definitively structured, thus leading to greater variability.

Starting from these theoretical frameworks which highlight both the role of individual characteristics and the importance of the open-skills component in determining the result in team sports, we aimed at finding a way to assess individual performance in volleyball in an ecological way. This is functional to study the role of psychological individual differences (e.g., mental aspects as general cognitive abilities, emotional control or coping mechanisms) in sport performance, which is a topic of interest for a growing body of research (see References [13,14,15,16] for reviews). This is particularly true for youth sport when the athletes are less specialized and, therefore, scouting only certain fundamentals (e.g., in volleyball, the attack percentage) would only provide a partial image of the athletes’ abilities. Despite being relatively small, the research field that takes a developmental psychological approach to the study of sport performance in developing athletes [17,18,19,20] underlines the importance of understanding cognitive functioning and other individual differences in their development, in order to identify actual predictors of sport performance. Indeed, testing developing athletes can provide insights into the relation of individual differences and motor learning or sports performance. Moreover, this is useful for talent identification as well. Previous research on talent identification in team sports concluded that the success in these sports depends not only on physical and technical skills, but leans on multidimensional factors, including psychological performance characteristics [21]. During growth, talented players need to monitor and improve themselves in each of these aspects in order to reach the top. Having a tool that allows keeping track of changings in the individual performance therefore becomes crucial. In this sense, devising an assessment measure that takes into account both all of the gestures performed by an athlete during competition and the unpredictability component given by the nature of team sports appears to be particularly meaningful.

Therefore, the rationale of the present study is devising and calculating an individual assessment measure that takes into consideration the performance of a volleyball player in a global manner, and verifying its interrater reliability and validity. Our hypothesis is that a measure developed according to the aforementioned criteria has good reliability and that its validity can be verified with respect to the players’ amount of experience (assuming that those who are less experienced tend to be less skilled players).

## 2. Methods

### 2.1. Participants

A total sample of 114 youth female volleyball players, aged between 11 and 17 years (mean age = 14 years 4 months, SD = 1 year 8 months), participated in this research. Because volleyball is a predominantly female sport in Italy, we enrolled only female athletes, to avoid an unequal distribution of gender in the sample. Participants were recruited from seven different academies of five cities in Northern Italy (Genova, Massa Carrara, Pietrasanta, Imperia, Verona). For all the participants, informed consent was signed by their parents. We included participants under 18 years of age with the aim of studying a developmental sample. To ensure that all participants experienced a reasonable amount of practice and competitions, we also enrolled athletes with at least three years of experience in volleyball (M = 5.66, SD = 2.36). We set the lower age bound at 11 years to include categories in which the game is sufficiently demanding on a competitive level.

### 2.2. Materials and Methods

With the aim to derive an individual measure from team performance, a scoring system similar to the one used in scouting was created. However, differently from the classic scouting system, we did not compute the percentage on each fundamental for each participant, but instead we scored every single contact the athlete made with the ball, in order to have a complete picture of the athlete’s contribution to the match. This is because, as stated above, in youth volleyball, the athletes, even when already playing in certain roles and positions, are generally less specialized and, therefore, their global performance would hardly be captured by the traditional scouting system. Moreover, this assessment method allows measuring on the same scale all the players, regardless of their roles, which would be impossible to achieve using a fundamental-based scoring method.

Therefore, with the aim of having a sufficient number of observations for each participant, we video-recorded at least three matches for each athlete during the 2017-2018 regional championship. All games were video-recorded by putting the camera behind the team on which the participants played, in a fashion the players were recognizable and the game actions clearly distinguishable. All videos (namely a total amount of 55 games and approximately 3895 min of game recorded), were subsequently scored by two blinded raters.

The raters were two expert volleyball coaches, who therefore provided a competent evaluation. They independently evaluated each athlete’s performance and calculated an individual performance index. To do so, every time that a participant (identified through her jersey number) touched (i.e., made contact with) the ball, her gesture was evaluated with the attribution of 0 points, 0.5 points, or 1 point. A score was attributed for every single contact a participant made with the ball, except for the block, which was not scored unless it ended the action (point gained or lost for the team). This because the block is a fundamental that is learned and perfected later compared to other ones and, in youth volleyball, it is more unlikely to conclude the action during a game. Scoring each block attempt would have been penalizing, especially for the middle-blockers, who typically perform this gesture more often that the teammates.

For each Set of the game, the final raw individual score was computed on a grid (see Figure 1) by adding together all the points scored by the participant. 

The criteria by which the points were attributed by the raters were previously discussed and chosen by expert volleyball coaches and are summed up and explained in Table 1.

### 2.3. Procedure

Even though the purpose of this protocol was to obtain an individual score, we could not ignore that volleyball is a team sport, in which the individual performance is highly influenced by the team and the opponents’ behavior. With the intention of controlling for these variables, we also collected the final score for each Set, and calculated the difference by which the participant’s team won or lost. This variable represents the context of the game (i.e., a summary measure of the balance between team and opponent during each Set). Therefore, we used this team performance measure to create a team-controlled individual performance index, as described below.

Firstly, for each Set, we calculated the athlete’s individual ratio (IR) between the total amount of touches (T) and the individual score computed by adding all the points (IS). For example, in Figure 1: IR=IS/T=9/12. Secondly, we used the team performance index (i.e., the difference by which the participant’s team won or lost; in Figure 1: TPI=25−18=+7) in order to calculate a weighted game index (WG) of all the Sets the athlete played in. This index represents the team performance weighted for the number of touches in each Set (T_i_, i.e., the “contribution” the athlete gave to the team) the athlete performed in all the Sets she played, and it is computed as follows:$$WG=\frac{\left(T_{1}*TPI_{1}+\;T_{2}*TPI_{2}+\;T_{3}*TPI_{3}\ldots +\;T_{n}*TPI_{n}\right)}{TT}$$
Thirdly, the final weighted individual index of performance (WIP) was calculated for each participant by saving the residuals of the regression of IR on WG. This methodological procedure was meant to control the individual performance for the team performance. Specifically, it ensured that, if an athlete often played against strong teams, the regression residual of IR on WG was magnified with respect to IR, and conversely, if an athlete often played against weaker teams, it was reduced. Since each observer gave one’s evaluations independently, two WIP scores were calculated for each athlete. This enabled us to analyze the inter-rater reliability.

## 3. Results

With respect to the IRs, as a preliminary measure of the interrater agreement, we computed the absolute difference between the two observer on all the 931 “units of observation”, namely all the Sets all the participants competed in (*min* = −.00, *max* = −.38, *M* = .04, *SD* = .05). Among the 931 calculated IR-differences, the 29.32% (*N* = 273) was .00 (i.e., the two raters calculated exactly the same ratio), the 40.82% (*N* = 380) was a difference lower than .05, the 20.19% (*N* = 188) was represented by a difference between .05 and .10 and only the 9.67% (*N* = 90) of the paired units of observation presented a difference higher than .10.

With the aim to test the reliability of our assessment measure, we then analyzed the correlations between the raw scores. Both the IRs (*r* = .899, *p* < .001) and the WGs (*r* = .985, *p* < .001) from the two observers showed a very high correlation. Moreover, the IR and the WIP (i.e., the “pure” individual score and the one regressed on the team performance) were highly correlated for both Observer 1 and Observer 2 (respectively *r* = .88 and *r* = .89, both *p* < .001).

At this point, we assessed the inter-rater agreement between the two judges on the WIP. The Pearson correlation between observers’ WIPs was significant and high, namely *r* = .841, *p* < .001. Moreover, if we compare the absolute value of the two observers’ WIPs, the difference between the means was very low (for Observer 1, *M* = .070, for Observer 2, *M* = .069) and n.s. (*t* = .17, *p* = .86). Also, the effect size (Cohen’s *d* = .03) was negligible [22]. This is a further sign of agreement between the observers.

Finally, we computed each athlete’s performance measure by using the mean WIP between Observer 1 and Observer 2. This final index of Volley Performance is the mean of the Weighted Individual Performance (i.e., the residuals of the regression of the athlete’s individual ratio on the weighted game index) indices assigned by Observer 1 and Observer 2.

In order to further evaluate the validity of this measure, we calculated correlations (both Pearson zero-order and controlling for age) between the index of Volley Performance and the years of experience, assuming that more experience would have been related to better performance (see Table 2). The correlation were significant, even with age partialled out.

## 4. Discussion

In this study, we devised a measure to assess individual performance in youth volleyball, by scoring every single athletic gesture the athlete performs during a competition.

We defined a series of criteria with the intention to score the athletes’ gestures, and we submitted the video-recordings of our participants’ volleyball matches to two independent raters. The high correlation of the IRs (*r* = .899) from the two observers show not only the goodness of the measure but also that isolating and quantifying a single athlete’s performance is possible by using feasible and expertise-built on criteria. Moreover, aware of aiming to derive an individual performance measure from a team sport context, we devised and calculated a control index that takes into account the context of the games. This index, calculated through the score difference in each Set played by the athlete and weighed on the actual participation of the athlete herself to the game, has allowed us to calculate an individual performance measure (i.e., the residuals of the regression of the “individual index” on the “game context” one). Using regression residuals is clearly a strong way to control for potentially confounding variables. This measure proved to be reliable (*r* = .841), showing a high degree of agreement between the two independent observers. The WIP reliability further supports the idea that that not only an individual measure of performance in team sports is valid, but also that it can be built by taking into account both the theoretical frameworks that value individual abilities and those which focus on the interaction/environmental aspects of team sports.

As stated above, this is useful in order to study the role of individual differences (e.g., general cognitive abilities, or emotional correlates of performance) in sports performance.

Indeed, as mentioned above, a growing body of research is becoming interested in the role of psychological differences in sport performance. For this reason, having also in team sports a measure of individual performance that could be correlated with those psychological variables is fundamental.

We created this measure based on a very ecological approach to the study of sport performance. Indeed, we did not design experimental settings with the purpose of recreating and isolating an individual measure of performance in an artificial fashion, but we took into account the actual competitions the athletes participated in during their sport seasons.

In individual sports, actual competitions scores are usually easier to gather, since there is a limited variance of error derived from other environmental aspects. Conversely, in team sports, research that studies individual differences rarely focuses on the actual “on-field” performance, mainly using “off-line” decision-making paradigms [23], which are of course valid outcome measures, but do not completely mirror the real “on-line” game situation [e.g., reference 20].

Having a “real” measure is nevertheless fundamental to properly study the developmental predictors of successful performance in sports, also with respect to talent development. Analyzing and monitoring the multidimensional aspects that influence talented young athletes during their growth will indeed provide more realistic goals for coaches to set and to assess during the season. This will also allow taking into account both physical and psychological aspects and the objective measures of game performance, therefore offering a global image of the athlete [21]. Moreover, this measure, assessing all the players on the same scale, independently from their roles, can also be applied to a whole team.

The creation of this online individual measure of performance in volleyball is, therefore, a significant example of how an expertise-based methodological approach can be used in order to acquire a performance measure that is as much as possible “cleaned up” from error variance while maintaining an ecological approach. Moreover, this measure is particularly suitable for youth sport because, differently from the classical scouting system, it takes into account all the gestures the players perform during the entire game.

Despite the methodological contribution offered, this study is not free of limitations. Firstly, we only included female participants, therefore, the indices we collected are not validated on a male sample. However, we think that the same protocol could also be applied to men’s youth volleyball. A limitation of the protocol is that, in order to control for WG, quite a large number of competitions of different teams is needed. Lastly, this protocol is quite expensive in terms of time because it requires the observers to watch and score every single contact every player makes with the ball, which is a significant amount of work. On the other hand, this way appears to be the most feasible method for youth athletes, whose performance needs to be assessed in a holistic manner, for the reasons explained above.

## 5. Conclusions

The main aim of this study was to devise a feasible measure to evaluate individual performance in youth volleyball. In doing so, we wanted it to be both ecological, namely based on actual competitions, and controlled for the “open” component characterizing team sports. This kind of measure could be easily used by future research aimed at analyzing individual differences in team sports, especially with a developmental outlook. This will allow identifying the best motor, psychological, and social correlates of optimal performance in volleyball, potentially offering indications to teams and coaches on which skills need to be trained more in order to succeed.

Not only being able to identify a clear relationship between individual abilities and performance would be useful to train those abilities, but also with respect to early identification and growing of talented players. In this sense, this method could be used by sports scientists working together with trainers, coaches, and scouts to underline key elements of the talent identification and development process [21] and could be potentially used to monitor the process of athletes’ improvement.

In conclusion, we believe that, despite its complexity, this method represents an effective way to measure and keep track of individual performance in volleyball and that the same procedure can also be applied to develop similar measures in other team sports, thus being a useful tool for developmental studies in the sports domain and, on the applicative side, for talent identification and development.

## Figures and Tables

**Figure 1 jfmk-04-00053-f001:**
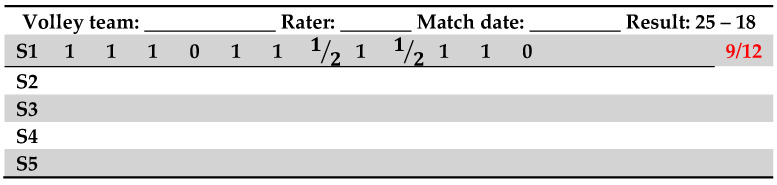
A prototype of observers’ grid.

**Table 1 jfmk-04-00053-t001:** Attribution criteria for the raw individual score calculation.

Points	Criterion	Examples
0	The athlete causes the loss of the point for her team	The ball is lost; Foul.
0.5	The athlete plays a ball which can still be played, but is not precise	A serve reaches the opponents’ court after touching the tape; The defense is played more than 1.5 m away from the setter (or the auxiliary setter), but the ball can still be played; The setter sets a ball that is hard-to-attack for the hitter; An attack reaches the opponents’ court after touching the tape. An attack is “wasted” by setting an “easy” ball to the opponents’ court, instead of spiking.
1	The athlete makes positive contact with the ball	Any gesture resulting in a point gained for the team; A serve^1^ reaches the opponents’ court without touching the tape^2^/Ace (namely a serve that results directly in a point); A pass^3^ is played less than 1.5 m away from the setter (or the auxiliary setter^4^); A proper (i.e., easy to attack) set^5^ played by the setter; An attack reaches the opponents’ court without touching the net; A challenging dig^6^, even when not precise; A block^7^, if resulting in a point gained for the team.

^1^ Serve: One of the six basic skills; used to put the ball into play. It is the only skill controlled exclusively by one player. ^2^ Tape: The top of the net. ^3^ Pass: Receiving a serve or the first contact of the ball with the intent to control the ball to another player. ^4^ Auxiliary setter: the player assigned to set when the designated setter cannot; usually the right-front player. ^5^ Set: The tactical skill in which a ball is directed to a point where a player can spike it into the opponent’s court. In this article, it will be written in all lower case in order to disambiguate the term that refers to a fraction of the game. ^6^ Dig: Passing a spiked or rapidly hit ball. Slang for the art of retrieving an attacked ball close to the floor. ^7^ Block: A defensive play by one or more front row players that is intended to intercept a spiked ball. It involves the combination of one, two, or three players jumping in front of the opposing spiker and contacting the spiked ball with their hands.

**Table 2 jfmk-04-00053-t002:** Correlations between volley performance and years of experience controlled for age.

	Volley Performance	Years of Experience	Age
Volley performance	1	−.203*	−.116
Years of experience	−.173*	1	−.720***

Note: Zero-order (Pearson) correlation above diagonal. Partial correlations controlled for age below diagonal. * *p* < −.05, *** *p* < −.001.

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
