# Peer review of "Assessing Individual Performance in Team Sports: A New Method Developed in Youth Volleyball"

_jfmk, 2019, doi:10.3390/jfmk4030053_

Round 1

Reviewer 1 Report

The paper “Assessing individual performance in team sports: a new method developed in youth volleyball” aims to perform an individual assessment in volleyball. Main data suggests a high level of agreement and a significant correlation with the amount of experience in volleyball players. The paper is interesting and within the scope of the journal. However, some concerns should be addressed:

1. Abstract. Please rewrite the abstract presenting clear data and sentences. It is not so easy to understand in this form.

2. Introduction. This section is well described and the state of the art is good. However, some important papers under this issue should be included highlighting some more findings regarding this issue. Please include references from different approaches regarding this problem: for instance, 1) Motricidade, 5(3), 7-11; 2) J Sci Med Sport 15(5): 457-462, 2012.

3. Introduction. Please highlight the rational of the study and the hypothesis.

4. Methods. Sample. Please explain the inclusion and exclusion criteria.

5. Methods. Please present the rational supporting the chosen protocols, explaining how the scientific and personal technical experience leaded to this analysis.

6. Statistics. Please include some reference for the Cohen´s d range.

7. Results. Good and well-writhen section.

8. Discussion. Please develop the main ideas and discuss them with scientific data namely presented in the introduction section.

9. Please include more clearly the limitations of the paper.

10. Please include the conclusion in a practical manner, underlined the implications for coaches and teams.

Author Response

Dear Reviewer,

thank you for your rapid and positive feedback. We have revised our manuscript following all of your suggestions. We list here in detail our responses to your comments and how we modified our article following your suggestions.

Abstract. Please rewrite the abstract presenting clear data and sentences. It is not so easy to understand in this form.

Thanks for the suggestion. We re-wrote the abstract trying to make the sentences simpler and to add some data. The new abstract starts at line 11 of the revised manuscript.

Introduction. This section is well described and the state of the art is good. However, some important papers under this issue should be included highlighting some more findings regarding this issue. Please include references from different approaches regarding this problem: for instance, 1) Motricidade, 5(3), 7-11; 2) J Sci Med Sport 15(5): 457-462, 2012.

Thank you for your positive feedback. As kindly suggested, we commented on different perspectives from which individual performance in volleyball can be analysed (see line 38-49 of the revised manuscript). We also included new references with respect to these findings [5-7], namely:

Eom, H. J.; Schutz, R. W. Statistical Analyses of Volleyball Team Performance. Research Quarterly for Exercise and Sport 1992, 63(1), 11-18. Marques, M. C.; Marinho, D. A. Physical parameters and performance values in starters and non-starters volleyball players: A brief research note. Motricidade 2009, 5(3), 7-11. Voelzke, M.; Stutzig, N.; Thorhauer, H. A.; Granacher, U. Promoting lower extremity strength in elite volleyball players: effects of two combined training methods. Journal of Science and Medicine in Sport 2012 15(5), 457-462.

Introduction. Please highlight the rationale of the study and the hypothesis.

We hereby copy lines 99-104 of the new manuscript:

Therefore, the rationale of the present study is devising and calculating an individual assessment measure that takes into consideration the performance of a volleyball player in a global manner and verifying its interrater reliability and validity. Our hypothesis is that a measure developed according to the aforementioned criteria has good reliability and that its validity can be verified with respect to the players’ amount of experience (assuming that those who are less experienced tend to be less skilled players).

Methods. Sample. Please explain the inclusion and exclusion criteria.

Thanks. We added a couple of sentences to explain the criteria. We hereby copy the new “Participants” section:

A total sample of 114 youth female volleyball players, aged between 11 and 17 years (mean age = 14 years 4 months, SD = 1 year 8 months), participated in this research. Because volleyball is a predominantly female sport in Italy, we enrolled only female athletes, to avoid an unequal distribution of gender in the sample. Participants were recruited from seven different academies of five cities in Northern Italy (Genova, Massa Carrara, Pietrasanta, Imperia, Verona). For all the participants, informed consent was signed by their parents. We included participants under 18 years of age with the aim of studying a developmental sample. To ensure that all participants experienced a reasonable amount of practice and competitions, we also enrolled athletes with at least three years of experience in volleyball (M = 5.66, SD = 2.36). We set the lower age bound at 11 years to include categories in which the game is sufficiently demanding on a competitive level.

Methods. Please present the rationale supporting the chosen protocols, explaining how the scientific and personal technical experience leaded to this analysis.

We inserted explanations to justify the analysis in the “Materials and Methods” section (see lines 122-127, 134-135, and 140-143 of the new manuscript).

Statistics. Please include some reference for the Cohen´s d range.

Thank you for the suggestion. We calculated Cohen’s d for the difference between means. We inserted a reference with respect to it, as requested. Line 209: “Also, the effect size (Cohen’s d = .03) was negligible [22].

Results. Good and well-writhen section.

Thank you for your kind feedback.

Discussion. Please develop the main ideas and discuss them with scientific data namely presented in the introduction section.

Thank you for the suggestion. We enriched the discussion and discussed our findings as suggested. (see lines 227-229, 237-246, and 256-263 of the new manuscript).

Please include more clearly the limitations of the paper.

We hereby copy lines 270-278 of the new manuscript:

Despite the methodological contribution offered, this study is not free of limitations.  Firstly, we only included female participants, therefore, the indices we collected are not validated on a male sample. However, we think that the same protocol could also be applied to men's youth volleyball. A limitation of the protocol is that, in order to control for WG, a quite high amount of competitions of different teams are needed. Lastly, this protocol is quite expensive in terms of time because it requires the observers to watch and score every single contact every player makes with the ball, which is a significant amount of work. On the other hand, this way appears to be the most feasible with youth athletes, whose performance needs to be assessed in a holistic manner, for the reasons explained above.

Please include the conclusion in a practical manner, underlined the implications for coaches and teams.

As suggested, we created a “Conclusion” section, in which we related to the objectives and we discussed the practical implications of our work – which we also enriched in the introduction (lines 90-95).

We hereby copy lines 279-297 of the new manuscript:

The main aim of this study was to devise a feasible measure to evaluate individual performance in youth volleyball. In doing so, we wanted it to be both ecological, namely based on actual competitions, and controlled for the “open” component characterizing team sports. This kind of measure could be easily used by future research aimed at analyzing individual differences in team sports, especially with a developmental outlook. This will allow identifying the best motor, psychological, and social correlates of optimal performance in volleyball, potentially offering indications to teams and coaches on which skills need to be trained more in order to succeed.

Not only being able to identify a clear relationship between individual abilities and performance would be useful to train those abilities, but also with respect to early identification and growing of talented players. In this sense, this method could be used by sports scientists working together with trainers, coaches, and scouts to underline key elements of the talent identification and development process [21] and could be potentially used to monitor the process of athletes’ improvement.

In conclusion, we believe that, despite its complexity, this method represents an effective way to measure and keep track of individual performance in volleyball and that the same procedure can also be applied to develop similar measures in other team sports, thus being a useful tool for developmental studies in the sports domain and, on the applicative side, for talent identification and development.

Reviewer 2 Report

Congratulations, it's an interesting manuscript.

However, it is important that the introduction also has a theoretical reference model and that the conclusions are clearer and according to the objectives proposed in the research.

Author Response

Dear Reviewer,

thank you for your rapid and positive feedback. We have revised our manuscript following your suggestions.

Congratulations, it's an interesting manuscript.

Thank you for your kind feedback.

However, it is important that the introduction also has a theoretical reference model…

Thank you. We better explained and declared our theoretical reference model (see lines 55-60, 77-80 of the new manuscript). Moreover, we commented on different perspectives from which individual performance in volleyball can be analysed (see line 38-49 of the revised manuscript). We also included new references with respect to these findings [5-7], namely:

Eom, H. J.; Schutz, R. W. Statistical Analyses of Volleyball Team Performance. Research Quarterly for Exercise and Sport 1992, 63(1), 11-18. Marques, M. C.; Marinho, D. A. Physical parameters and performance values in starters and non-starters volleyball players: A brief research note. Motricidade 2009, 5(3), 7-11. Voelzke, M.; Stutzig, N.; Thorhauer, H. A.; Granacher, U. Promoting lower extremity strength in elite volleyball players: effects of two combined training methods. Journal of Science and Medicine in Sport 2012 15(5), 457-462.

...and that the conclusions are clearer and according to the objectives proposed in the research.

As suggested, we created a “Conclusion” section, in which we related to the objectives and we discussed the practical implications of our work – which we also enriched in the introduction (lines 90-95). Moreover, we added many references to our objectives in the Discussion section (lines 225-246, 256-256).

We hereby copy lines 279-297 of the new manuscript:

The main aim of this study was to devise a feasible measure to evaluate individual performance in youth volleyball. In doing so, we wanted it to be both ecological, namely based on actual competitions, and controlled for the “open” component characterizing team sports. This kind of measure could be easily used by future research aimed at analyzing individual differences in team sports, especially with a developmental outlook. This will allow identifying the best motor, psychological, and social correlates of optimal performance in volleyball, potentially offering indications to teams and coaches on which skills need to be trained more in order to succeed.

Not only being able to identify a clear relationship between individual abilities and performance would be useful to train those abilities, but also with respect to early identification and growing of talented players. In this sense, this method could be used by sports scientists working together with trainers, coaches, and scouts to underline key elements of the talent identification and development process [21] and could be potentially used to monitor the process of athletes’ improvement.

In conclusion, we believe that, despite its complexity, this method represents an effective way to measure and keep track of individual performance in volleyball and that the same procedure can also be applied to develop similar measures in other team sports, thus being a useful tool for developmental studies in the sports domain and, on the applicative side, for talent identification and development.